# Melatonin in Newborn Infants Undergoing Surgery: A Pilot Study on Its Effects on Postoperative Oxidative Stress

**DOI:** 10.3390/antiox12030563

**Published:** 2023-02-24

**Authors:** Serafina Perrone, Carmelo Romeo, Lucia Marseglia, Sara Manti, Cristina Rizzo, Silvia Carloni, Maria Cristina Albertini, Walter Balduini, Giuseppe Buonocore, Michael D. Weiss, Eloisa Gitto

**Affiliations:** 1Department of Medicine and Surgery, Neonatology Unit, Pietro Barilla Children’s Hospital, University of Parma, 43121 Parma, Italy; 2Department of Human Pathology of the Adult and Developmental Age, Pediatric Surgery Unit, University of Messina, 98124 Messina, Italy; 3Department of Human Pathology of the Adult and Developmental Age, Neonatal Intensive Care Unit, University of Messina, 98125 Messina, Italy; 4Department of Biomolecular Sciences, University of Urbino Carlo Bo, 61029 Urbino, Italy; 5Department of Molecular and Developmental Medicine, University of Siena, 53100 Siena, Italy; 6Department of Pediatrics, University of Florida, Gainesville, FL 32610, USA

**Keywords:** surgical patients, perioperative oxidative stress, isoprostanes, lipid peroxidation, protein peroxidation, free iron, AOPP

## Abstract

Surgery is frequently associated with excessive oxidative stress. Melatonin acts as an antioxidant and transient melatonin deficiency has been described in neonatal surgical patients. This randomized, blinded, prospective pilot study tested the hypothesis that oral melatonin supplementation in newborn infants undergoing surgery is effective in reducing perioperative oxidative stress. A total of twenty-three newborn infants requiring surgery were enrolled: 10 received a single dose of oral melatonin 0.5 mg/kg in the morning, before surgery (MEL group), and 13 newborns served as the control group (untreated group). Plasma concentrations of melatonin, Non-Protein-Bound Iron (NPBI), Advanced Oxidation Protein Products (AOPP), and F2-Isoprostanes (F2-IsoPs) were measured. Both in the pre- and postoperative period, melatonin concentrations were significantly higher in the MEL group than in the untreated group (preoperative: 1265.50 ± 717.03 vs. 23.23 ± 17.71 pg/mL, *p* < 0.0001; postoperative: 1465.20 ± 538.38 vs. 56.47 ± 37.18 pg/mL, *p* < 0.0001). Melatonin significantly increased from the pre- to postoperative period in the untreated group (23.23 ± 17.71 vs. 56.47 ± 37.18 pg/mL; pg/mL *p* = 0.006). In the MEL group, the mean blood concentrations of NPBI, F2-IsoPs, and AOPP significantly decreased from the pre- to the postoperative period (4.69 ± 3.85 vs. 1.65 ± 1.18 micromol/dL, *p* = 0.049; 128.40 ± 92.30 vs. 50.25 ± 47.47 pg/mL, *p* = 0.037 and 65.18 ± 15.50 vs. 43.98 ± 17.92 micromol/dL, *p* = 0.022, respectively). Melatonin concentration increases physiologically from the pre- to the postoperative period, suggesting a defensive physiologic response to counteract oxidative stress. The administration of exogenous melatonin in newborn infants undergoing surgery reduces lipid and protein peroxidation in the postoperative period, showing a potential role in protecting babies from the deleterious consequences of oxidative stress.

## 1. Introduction

Surgery is frequently associated with excessive oxidative stress. The surgical procedure promotes a local and systemic inflammatory response with immunological, biohumoral, and endocrine–metabolic consequences [1,2,3,4]. The release of cytokines induces amplification of the inflammatory response with the activation of macrophages, mast cells, and platelets which, in turn, determine the formation of free radicals [5,6,7]. Pain in relation to surgery is also a problem. Pain represents another source of free radical burden in human patients [8,9].

Melatonin acts as an antioxidant [10,11]. This endolamine possesses both direct antioxidant effects by acting as a free radical scavenger (singlet oxygen (^1^O_2_), superoxide anion radical (•O_2_^−^), hydroperoxide (H_2_O_2_), hydroxyl radical (•OH), and the lipid peroxide radical (LOO•), and indirect antioxidant effects by increasing the efficiency of mitochondrial electron transport and by activating the most important antioxidant enzymes: superoxide dismutase (SOD), catalase (CAT), and glutathione peroxidase (GPx). The therapeutic potential of melatonin as an agent to counteract the consequences of free radicals has been demonstrated in newborn infants [12,13,14,15,16]. As an antioxidant and anti-inflammatory substance, melatonin serves to combat several comorbidities, such as metabolic syndrome, ischemic diseases, and sepsis, which aggravate the conditions needing surgery. The response to trauma induced by surgical procedures involves an acute inflammatory phase to initiate the healing process, but an excessive cytokine storm can lead to a systemic inflammatory response syndrome which, in turn, may lead to oxidative stress and mortality [17,18]. The physiological concentration of melatonin varies among newborn infants undergoing surgery, and transient melatonin deficiency has been described in neonatal surgical patients [19,20,21].

A multifactorial analysis in the paper by Dragoumi et al. revealed a statistically significant difference between 24 h postoperative melatonin values in age groups three (3–6 years) and five (6–12 years old). Age group three (3–6 years) showed significantly lower postoperative melatonin values compared with age group five (6–12 years) [22].

Neonates produce low levels of pineal melatonin and have deficiencies in diurnal variation during the first week of life. The secretion of melatonin increases rapidly to reach 50% of the adult value at the age of 1 year in preterm and term infants.

Low perioperative plasma concentrations of melatonin in critically ill patients have been associated with delirium [23]. Nocturnal concentrations of melatonin were significantly lower on the first than on the second or third night after surgery [24].

Because of the occurrence of oxidative stress in patients undergoing surgery, the putative application of melatonin as a therapy has emerged, based on its cytoprotective properties [25,26].

The current literature demonstrates a great heterogeneity of experimental conditions, including animal models, oxidative stress markers, methods, and time periods of measurement. Published data in humans do not allow a reliable conclusion on the effect of melatonin supplementation on oxidative stress in patients requiring surgery. The majority of studies in human subjects focus on the effect of melatonin in adult patients, without taking into consideration the unique physiology present during the neonatal period.

We previously demonstrated that melatonin administration in preterm newborns reduced lipid peroxidation in the first days of life, showing a potential role in protecting high-risk newborns. We observed a lower concentration of isoprostanes, a reliable marker of arachinoid acid oxidation, in treated newborns than in controls [27].

Moreover, our research group for the first time reported the pharmacokinetic profile of 0.5 mg/kg melatonin given orally over 4 h. This dosage was given to five neonates with hypoxic–ischemic encephalopathy undergoing therapeutic hypothermia. The highest plasma concentration occurred between 3 and 12 h after the completion of the infusion, which was a longer half-life in comparison with animals and human adults [28].

### Objective

The aim of this pilot research was to evaluate the effectiveness of melatonin supplementation on perioperative oxidative stress in newborn infants undergoing surgery. We hypothesized that newborns who receive melatonin before surgery have lower oxidative stress biomarkers than those who do not receive melatonin.

## 2. Materials and Methods

An evaluator-blinded, randomized controlled trial of melatonin versus placebo for newborns undergoing surgery was performed and is registered on the Clinical Trials Registry (Registration Number: NCT04785183). Thus, this study was part of a larger clinical trial (NCT04785183). As there were no available data, no power calculation was performed for this RCT study; the study sample size was estimated from an a priori calculation based on best available evidence in newborns where pilot RCTs have been published [28,29,30,31,32].

The study was conducted at the Neonatology Unit of the AOU “G. Martino” in Messina. The local Ethics Committee approved the specific amendment to the study protocol (n. 125-22, 2022).

Written, informed consent was obtained from parents. The inclusion criteria were gestational age >34 weeks and the need for surgery in the neonatal period. Exclusion criteria were sepsis, inborn errors of metabolism, newborns suffering from perinatal hypoxia, newborns of mothers with mental disorders, or surgical intervention in the afternoon or in the night to eliminate conditions that could affect melatonin production.

Newborns were also excluded in cases of withdrawal of informed consent, insufficient blood sampling, and hemolysis of the blood sample, as hemolysis interferes with the biochemical determination of oxidative stress biomarkers.

### 2.1. Recruitment and Randomization

If infants were identified as potentially eligible, the caregivers were provided with information about the study. If the caregiver agreed, an initial assessment was then conducted to confirm eligibility and obtain consent to participate in the study. Eligible infants with consent were then randomly allocated to either the Melatonin or the Untreated group (Figure 1).

Permuted, blocked randomization was conducted by a biostatistician using com puter-based sequences, and group allocation was provided in concealed opaque envelopes. Participants’ parents were informed of their infant’s group allocation.

### 2.2. Interventions

Newborns in the MEL group received a single dose of oral melatonin 0.5 mg/kg in the morning, before surgery.

Melatonin (Pisolino^®^ Gocce, Pediatrica, Italy) was administered by a nurse through a nasogastric tube. The product is present in the register of food supplements of the Ministry of Health website (http://www.ministerosalute.it/alimenti/ dietetica. Last Access: 1 January 2021) and classified with the following code: 62853.

This product is subject to the European Directive on foods according to the DL n. 169 of 21 May 2004 and not to the European Directive on Medicines 2001/20/EC implemented at Italian level with D.L. n. 211 of 24 June 2003. Melatonin administration has a good safety profile, with no known adverse effects [33].

Newborns in the untreated group received 0.5 mL of 5% glucose solution in the morning, before surgery, by a nurse through a nasogastric tube.

The clinical characteristics of the patient population are reported in Table 1.

Samples of 0.2 mL of plasma were collected at 1 h before surgery and at 1 h after surgery.

Plasma concentrations of melatonin, non-protein-bound iron (NPBI, μmol/dL), advanced oxidation protein products (AOPP, μmol/dL), and F2-isoprostanes (F2-IsoPs, pg/mL) were measured.

Clinical and research staff remained unaware of test group assignments until the completion of data analysis.

### 2.3. Perioperative Management of Newborn Infants, Analgesia and Anesthesia

Standard pre-oxygenation for 60 s was performed and, to achieve an end-tidal oxygen concentration of at least 90%, a tight-fitting mask and 6 L/min oxygen flow for 40 s were used. Either thiopental (5–10 mg/kg) or propofol (3.0–3.5 mg/kg) was administered as the induction agent. In hemodynamically unstable patients, ketamine (1–2 mg/kg) was the drug of choice for intravenous induction.

Endotracheal intubation was performed after a single dose of succinylcholine (3 mg/kg) to facilitate airway management.

The use of the opioid fentanyl (1–5 µg/kg) served for the maintenance of anesthesia during surgery. Oxygen was titrated to achieve a peripheral oxygen saturation of at least 92% during surgery (Sat O_2_ range: 86–96%).

### 2.4. Melatonin and Oxidative Stress Biomarker Measurements

Peripheral blood (0.5 mL) was collected from each newborn infant at the preoperative period (T0) and at the postoperative period (T1). The samples were immediately centrifuged (RTM 1500, T 4 °C, 10 min) to remove cells and obtain the supernatant, which was then separated into three different microtest tubes, one of which contained BHT (butylated hydroxytoluene), and stored at −80 °C. The obtained samples were subsequently analyzed to measure melatonin concentration and OS biomarkers.

Plasma melatonin concentrations were measured using high-performance liquid chromatography coupled with tandem mass spectrometry (LC-MS/MS) (API 4000 triple-quadrupole Tandem Mass Spectrometer coupled with HPLC Agilent 1200 series system) according to the method of Wang et al. [34]. This ultra-high sensitive bioanalytical method for plasma melatonin used water as calibration matrix. This method allowed measurement accuracy to be achieved using a very small sample volume.

The injection volume was 10 µL and the injector needle was washed with 100 µL of both methanol–water (20:80) and methanol–water (80:20), once before injection and 5 times after injection. The aliquots of samples were gradient-eluted at 0.5 mL/min using (A) 2 mM ammonium formate and 0.1% formic acid in water and (B) acetonitrile (60% A for 1 min, 60–25% A for 4 min, 25–60% A for 3 min, and post run for 3 min) with a total run time of 11 min. Chromatography was performed on column C8. The lower limit of quantitation of the method was verified to be 1.0 pg/mL and the method exhibited a linear range of 1–5000 pg/mL.

AOPP and F2-IsoPs were detected as markers of protein and lipid OS-induced injury, respectively, using spectrophotometry on a microplate reader and by the LC-MS/MS methodology as previously described [35,36]. A spectrophotometric assay based on reaction of chlorinated protein with potassium iodide in an acid solution was used for AOPP quantification. Sixty microliters of the plasma sample was diluted 1:10 in chloramine-T standard solution, then 30 mL of potassium iodide and 60 mL of acetic acid were added before reading the absorbance at 340 nm. AOPP concentrations were expressed as micromolar chloramine-T equivalents.

The API 4000 tandem mass spectrometer coupled with HPLC Agilent 1200 series, included a degasser, a binary pump, a thermostated column compartment, and a well-plate autosampler. It was used to measure isoprostanes. Chromatography separation was carried out at a temperature of 30 °C using a mixture of an aqueous solution of acetic acid (Eluent 1) and acetonitrile (Eluent 2). For measurements, the tandem mass spectrometer ran in multiple reaction monitoring, with the electroscopy source operating in negative ion mode and exploiting the transitions m/z353:3 > 193:2 for F2 IsoPs and 357:3 > 197:2 for the internal standard d4-8-iso-PGF2α. An Agilent MassHunter workstation was used for the control of equipment, data acquisition, and analysis.

NPBI was detected as a marker of OS potential risk with an HPLC-DAD system (Agilent 1100 series) using the method described by Paffetti et al. [37]. The method is based on preferential chelation of NPBI by a large excess of the low-affinity ligand disodium nitrilotriacetic acid (NTA). Ten milliliters of NTA (0.8 M, pH 7.0) was added to 100 μL of the sample, mixed, and allowed to stand for 30 min at room temperature. It was then diluted with an equal volume of 5 mmol/L MOPS pH7, mixed, and left to stand for 20 min at room temperature. The solution was transferred to a 20 kDa Vecta Spin Micro-Whatman microcentrifuge filter and centrifuged at 13,660× *g* for 40 min at 4 °C. The filtrate was then analyzed. The analytic system detected iron as a ferric nitrate standard down to a concentration of 0.01 μM.

### 2.5. Statistics

Statistical analysis was carried out by a statistician who was aware of the study aim using SPSS version 25.0 for Windows (IBM, Armonk, NY, USA). Normal distribution of data was evaluated using the Kolmogorov–Smirnov test. Data were expressed as median, mean, and SD. Parametric data were analyzed with the Student *t* test while non-parametric data were evaluated with the chi-squared test. All data with *p* < 0.05 were considered statistically significant.

## 3. Results

### 3.1. Melatonin Pre- and Postoperative Plasma Concentrations

To investigate the variation of circulating melatonin in newborn infants who underwent surgery, melatonin plasma concentration was evaluated in pre- and postoperative patients.

In this study, 10 newborns received a single dose of oral melatonin 0.5 mg/kg, in the morning before surgery (MEL group) and 13 newborns served as the untreated control group. Before analyzing oxidative stress markers, we decided to evaluate the concentration of plasma melatonin before and after the newborns underwent surgery. As indicated in Figure 2, plasma melatonin concentrations were significantly higher in newborns treated with melatonin than in the untreated group, both in the preoperative period (Figure 2A; 1265.50 ± 717.03 pg/mL vs. 23.23 ± 17.71 pg/mL; *p* < 0.0001) and in the postoperative period (Figure 2B; 1465.20 ± 538.38 pg/mL vs. 56.47 ± 37.18 pg/mL; *p* < 0.0001).

Melatonin serum concentrations significantly increased in the postoperative period in the untreated group (23.23 ± 17.71 pg/mL vs. 56.47 ± 37.18 pg/mL; *p* = 0.006) (Figure 2A).

### 3.2. Pre- and Postoperative Blood Concentrations of Markers of Oxidative Stress

Blood concentration of NPBI, F2-IsoPs, and AOPP are reported in Table 2.

The mean blood concentration of NPBI, F2-IsoPs, and AOPP significantly decreased from the pre- to postoperative period in the Mel group (Figure 3).

As reported in Figure 3, in the untreated group a significant decrease in NPBI in the T1-postoperative condition was observed (Figure 3A), while levels of F2-IsoPs and AOPP showed no significant differences between the pre- and postoperative period (Figure 3C,E). In the Mel group, levels of NPBI, F2-IsoPs and AOPP significantly decreased from the pre- to postoperative period (Figure 3B, *p* = 0.049; Figure 3D, *p* = 0.037; Figure 3F, *p* = 0.022, respectively).

No side effects were reported during the study period.

## 4. Discussion

Several studies report a reduction in endogenous melatonin plasma concentration in patients undergoing major abdominal surgery, coronary artery bypass grafting, and gynecological surgery [20,38,39,40,41]. A reduction in endogenous melatonin secretion has been demonstrated at different times in relation to the specific surgical intervention.

Cronin et al. reported that nocturnal concentrations of melatonin were significantly (*p* = 0.005) lower on the first than on the second or third night after surgery. This finding raised the hypothesis that melatonin suppression and associated sleep disturbance might be prevented by melatonin replacement [41].

Guo et al. demonstrated a variation in indolamine levels during coronary artery bypass surgery, as well as in the immediate postoperative period [42]. This alteration appeared to be absent on the day following the surgery. It was suggested that there was a possible correlation between postoperative sleep disturbances and reduced secretion of melatonin on the day after the operation.

Alterations in plasma levels of melatonin, due to surgical stress, have been also demonstrated in the pediatric population [22] with a reduction in melatonin values in the postoperative time, at 24 h, that was significantly higher in patients aged between 3 and 6 years of age compared with patients at 6–12 year of age. The difference between neonatal and pediatric patients may be due to the irregular secretion of melatonin in the first years of life. An irregular synthesis of melatonin in the neonatal period and in the first months of life leads to a transient deficiency of melatonin [43,44].

The results from these studies lay the foundation for the study of endogenous melatonin serum concentrations and administration of exogenous melatonin to counteract and antagonize surgical stress, and its consequences. Pain in relation to surgery is a well-known problem [26,45,46]. Pain causes an increase in free radicals that cannot be properly neutralized by the antioxidant system in preterm newborns [47]. Chronic pain is induced and supported by proinflammatory cytokines, free radicals, and reactive oxygen species, creating a self-sustaining vicious circle.

Painful stimuli during early brain development can alter the brain microstructure and topographical organization of afferent thalamo-cortical projections, resulting in the development of the atypical somatosensory cortex [48]. The exposure to procedural pain of preterm newborn is associated with reduced width of the frontal and parietal lobes of the brain, reduced functional connectivity in the temporal lobes, and smaller volumes of subcortical brain structures, including the amygdala, thalamus, and basal ganglia [49]. There is also increasing concern about the potential adverse effects of perioperative anesthesia and anesthetic-related hemodynamic changes on the developing brain of newborns.

Melatonin may offer an atoxic adjunct therapy in ameliorating this condition in the pre- and postoperative period [50].

The analgesic power of melatonin seems to be linked both to its direct action on melatonergic, opioidergic, benzodiazepinergic, muscarinic, nicotinic, serotonergic, and adrenergic receptors, and indirectly to its inflammatory and antioxidant effects.

The analgesic activity of melatonin during endotracheal intubation and mechanical ventilation was previously reported by our research team. Two groups, each of 30 preterm newborns, treated with 10 mg/kg of intravenous melatonin + fentanyl or fentanyl alone were compared [50]. After intubation, the Premature Infant Pain Profile (PIPP) scale assessed at 12, 24, 48, and 72 h of invasive ventilation was significantly lower in the group of patients treated with melatonin and fentanyl compared with infants treated with standard care [50].

The data obtained in the current study showed a significant increase in the secretion of endogenous melatonin in patients undergoing major surgery. This is consistent with the results reported by Marseglia et al. showing that the epiphysis appears to be hyperstimulated in critically ill newborns requiring hospitalization in neonatal intensive care [51]. The rise in melatonin levels could represent a physiological attempt to actively respond to oxidative stress, secondary to a serious and critical disease process. Furthermore, this same physiological response seems to be compromised with advancing age, showing reduced melatonin concentrations in the adult patient in critical condition [25].

Our study confirmed the hypothesis that the increase in endogenous melatonin correlated to the physiological response of the organism to a stressful situation. The endogenous increase in melatonin in response to surgical trauma in the immediate postoperative phase, in newborn infants, is innovative and striking. However, the serum concentration of endogenous melatonin in the untreated neonates was not effective in reducing the concentrations of the biomarkers of oxidative stress examined.

Recently, our research team studied the peak plasma concentration and the long half-life from a single dose of melatonin (0.5 mg/kg or three boluses 0.5–1 mg/kg at 24 h intervals) after intragastric administration in neonates [31]. Since the endogenous increase in the concentrations of melatonin in neonates did not decrease all the biomarkers of oxidative stress following surgery, we examined, the use of exogenous melatonin as a preoperative “treatment” and examined if the markers of oxidative damage were reduced. Similar strategies have been attempted in the adult population. The prospective, randomized, double-blinded study of Kücükakin et al. reported that intraoperative administration of 10 mg of melatonin during laparoscopic cholecystectomy was not associated with a change in the serum concentration of inflammatory and pro-oxidant markers, in relation to the administration of the placebo [52]. In that study, the exact dose of exogenous melatonin to obtain an antioxidant effect in the patient was not known. A standard dose of 10 mg in an adult patient is lower than 0.5 mg/kg in the neonatal population. In our study, the administration of exogenous melatonin preoperatively resulted in a significant reduction in the levels of the biomarkers of oxidative damage (F2-Isoprostanes, AOPP, NPBI). Isoprostanes are a family of lipid mediators generated by the action of cyclooxygenase on long-chain unsaturated fatty acids. Prostanoids are more stable compared with other peroxidation products, such as aldehydes or peroxyl radicals; thus, they can be detected in biologic fluids. They are therefore reliable markers of in situ oxidative injury. NPBI is a low-molecular-mass iron form, free from binding to plasma proteins. Iron toxicity is directly proportional to the quantity of hydrogen peroxide required to produce hydroxyl radicals through the Fenton reaction. Furthermore, lipid exposure to high concentrations of NPBI leads to formation of IsoPs. Non-protein-bound iron is a marker of potential oxidative stress because it indicates increased susceptibility to oxidative damage, especially in in vivo studies. All the patients enrolled in the study who underwent surgery belonged to a very sensitive patient population in which the “stress condition” was marked. In fact, the levels of oxidative stress biomarkers were found to be higher, in both the Untreated and Mel-treated groups, than in those measured in healthy infants we had analyzed in a previous study [53]. Melatonin, on the other hand, at physiological plasma concentration, tended to reduce oxidative stress during the postoperative period, which could be recognized as a physiological attempt to respond to oxidative stress, but it reached a significant effect only after exogenous administration, which led to a more than 100-fold increase in melatonin plasma concentration.

The argument for “treatment” with melatonin is therefore supported by the insufficient capacity of endogenous melatonin to neutralize the burden of oxidative stress induced by surgical trauma.

This study has some limitations. First, this is a pilot study with a small sample size, in which randomization could play an important role.

Second, surgery was performed on newborns at different ages after birth, so there was a certain heterogeneity in the time periods of measurements. Nevertheless, the disparity between groups did not affect the power of the statistical analysis. Third, the enrolled population was affected by various pathologies requiring surgery due to the ‘rarity’ of these kinds of diseases in the neonatal population. We believe that research studies should be performed as homogeneously as possible, ideally with more patients and, possibly, with patients sharing the ‘same’ pathology.

The small sample size and pilot nature of this study is likely to have limited the levels of significance below what might have been established in a larger sample size; that is, recommendations for further larger-scale research are required.

The strengths of this work are worthy of consideration: the capacity for endogenous production of melatonin in response to surgical stimulus and the significant impact of exogenous melatonin in reducing all markers of oxidative stress. The accurate measurement of oxidative stress in vivo is necessary to investigate its role in diseases, or to evaluate the efficacy of treatment. Free radicals have a very short half-life (of the order of few seconds), and their measurement in vivo poses many challenges. However, oxyradical derivatives (e.g., hydrogen peroxide or lipid hydroperoxides) are stable and have long half-lives (hours to weeks) and thus may be measured and monitored repeatedly. F2 ispoprostanes, AOPP and NPBI are novel and reliable markers of oxidative stress [54,55]. Reference values for the neonatal population are now available [53]. These intervals are necessary for all clinical laboratory tests, and they are an important task for screening, diagnosis, and monitoring of perinatal diseases. Plasma AOPP levels provide information regarding aspects of protein involvement in free radical reactions, namely oxidized plasma proteins that have lost their oxidant properties. AOPP is a very important biomarker of oxidative stress because the proteins are the major targets of free radicals, being present in abundance in cells, plasma, and most tissues.

Future studies should focus on the role of exogenous melatonin, to establish whether a reduction in induced oxidative stress can have a clinical impact in reference to post-operative pain, prevention of post-surgical infectious complications, or development of a more rapid recovery nutrition, as well as a reduction in hospitalization times. Its multifactorial therapeutic potential is unique to melatonin and is not shared by any other drug candidate for the perioperative management of newborn infants.

## 5. Conclusions

Melatonin concentration increases physiologically from the pre- to postoperative period, suggesting a tentative ability to counteract oxidative stress. The administration of exogenous melatonin in newborn infants undergoing surgery reduces lipid and protein peroxidation in the postoperative period, showing a potential role in protecting babies from the deleterious consequences of postoperative oxidative stress.

## Figures and Tables

**Figure 1 antioxidants-12-00563-f001:**
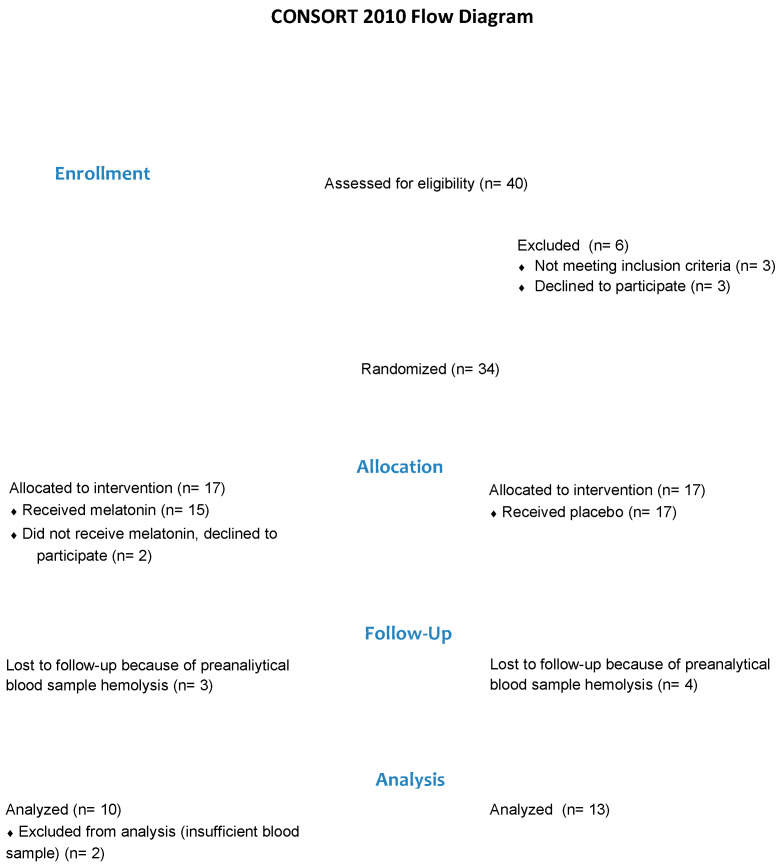
Flow of participants.

**Figure 2 antioxidants-12-00563-f002:**
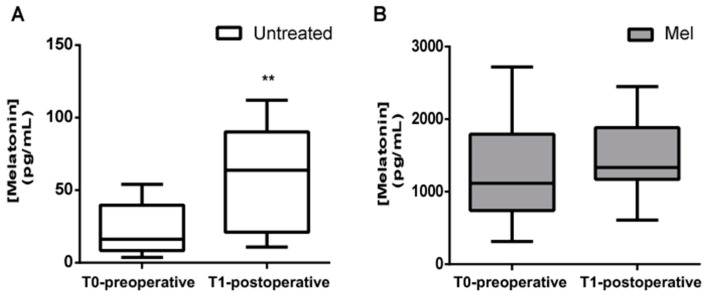
Preoperative and postoperative plasma melatonin concentrations (pg/mL). Plasma melatonin concentration evaluated before (T0-preoperative) and after (T1-postoperative) surgery in (**A**) untreated (Untreated) newborn infants and (**B**) newborns treated with Melatonin (Mel). Melatonin significantly increased in T1-postoperative period in the Untreated group; ** *p* ≤ 0.01.

**Figure 3 antioxidants-12-00563-f003:**
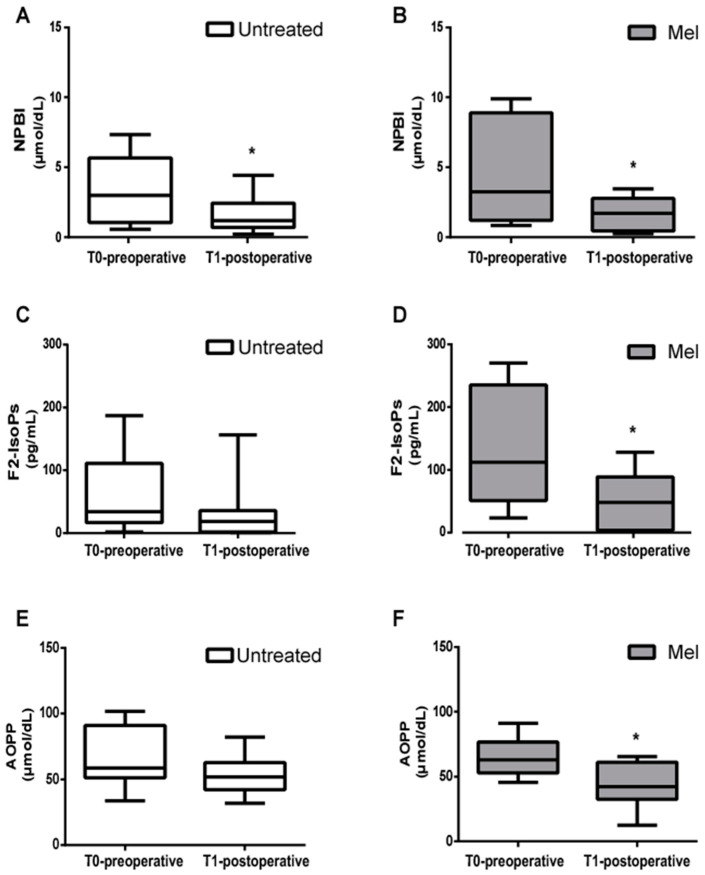
Biomarkers of oxidative stress damage. (**A**,**B**) Mean blood concentration of Non-Protein-Bound Iron (NPBI), (**C**,**D**) F2-Isoprostanes (F2-IsoPs), and (**E**,**F**) Advanced Oxidation Protein Products (AOPP), evaluated before (T0-preoperative) and after (T1-postoperative) surgery in untreated (Untreated) newborn infants and newborns treated with Melatonin (Mel). Oxidative stress markers were significantly decreased in T1-postoperative Mel-treated newborn infants; * *p* < 0.05.

**Table 1 antioxidants-12-00563-t001:** Clinical characteristics of the enrolled patients.

	MEL * Group (n = 10)	Untreated Group (n = 13)	*p*-Value
**Gestational Age (wks) ***	38.26 ± 3.40	37.13 ± 2.64	NS *
**Birth weight (g) ***	2920.10 ± 635.23	2865.35 ± 498.34	NS
**Gender (F;M) ***	6 (F); 4 (M)	8 (F); 5 (M)	NS
**Age at surgery (days of life)**	10 ± 7	7 ± 5	NS
**Average no. of minutes before surgery** **(time of sampling)**	240 ± 85	255 ± 45	NS
**Average time at intervention (melatonin or placebo, hours)**	8 ± 1.5	8 ± 2	NS
**Enterostomy**	4	6	--
**Omphalocele**	1	0	--
**Meconium ileus**	2	3	--
**Severe hydronephrosis**	1	2	--
**Sacrococcygeal teratoma**	0	1	--
**Intestinal duplication**	1	1	--

* wks: weeks; MEL: melatonin; NS: not significant; g: grams; F: female; M: male.

**Table 2 antioxidants-12-00563-t002:** Plasma levels of NPBI, F2-IsoPs, and AOPP in untreated newborn infants and newborns treated with melatonin (Mel) in the preoperative (T0) and postoperative (T1) period.

Biomarker	Untreated	Mel	*p*-Value
**NPBI T0**	3.24 (2.45); 2.99 (1.05–5.66)	4.69 (3.85); 3.24 (1.20–8.89)	ns
**NPBI T1**	1.70 (1.32); 1.19 (0.69–2.43) ^§^	1.65 (1.18); 1.71 (0.43–2.77) *	ns
**F2–IsoPs T0**	57.35 (59.92); 33.90 (16.60–110.80)	128.40 (92.30); 112.00 (51.00–235.22)	0.035
**F2–IsoPs T1**	29.47 (43.17); 18.30 (1.91–35.55)	50.25 (47.47); 47.75 (3.51–88.70) **	ns
**AOPP T0**	67.83 (23.18); 58.70 (51.28–90.78)	65.18 (15.50); 62.89 (52.89–76.50)	ns
**AOPP T1**	52.84 (14.13); 51.82 (42.17–62.83)	43.98 (17.92); 42.33 (32.52–61.01) ***	ns

Data are presented as mean (standard deviation); median (interquartile range). ^§^
*p* = 0.028, * *p* = 0.049; ** *p* = 0.037; *** *p* = 0.022.

## Data Availability

The data presented in this study are available on request from the corresponding author. The data are not publicly available due to privacy.

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
