# Peer review of "Melatonin in Newborn Infants Undergoing Surgery: A Pilot Study on Its Effects on Postoperative Oxidative Stress"

_antioxidants, 2023, doi:10.3390/antiox12030563_

Round 1
Reviewer 1 Report
This research group has a very well known and respected track record on perinatal oxidative stress research, with specific expertise on melatonin as modulator, and – associated with this – to improve the outcome. In the current paper, these authors report on a pilot type of study (suggest, perhaps been added to the title).
Control melatonin increases. This is somewhat unanticipated. Could this relate to the diurnal fluctuations, or not (chronopharmacology ?), besides the suggested stress response ?
Table 1: were data distributed normally (as mean and sd are provided), otherwise median and IQR or range are likely more accurate ?
It would be useful to have some more information on the oxygenation/saturation targets used during the surgery. Pre)-oxygenation has been applied, but how has oxygen been handled throughout surgery/anesthesia. Please check the fentanyl dose, should likely be microgr ?
Please be consistent on the names used, either T1 and T2, or T0 and T1 (text versus figures) a please provide more details to the timing of sample collection related to surgery, and to dosing ? There is neither information yet on the melatonin measurements (sampling, how it has been measured)
Editing issue on how the references are mentioned in the text ?
I therefore respectfully suggest 'major = moderate' revision
Author Response
Reviewer 1
This research group has a very well known and respected track record on perinatal oxidative stress research, with specific expertise on melatonin as modulator, and – associated with this – to improve the outcome. In the current paper, these authors report on a pilot type of study (suggest, perhaps been added to the title).
Control melatonin increases. This is somewhat unanticipated. Could this relate to the diurnal fluctuations, or not (chronopharmacology ?), besides the suggested stress response ?
Re: We thank the reviewer for the opportunity to ameliorate our paper. Both in the treated and untreated newborns, sampling was carried out in the morning, in the early phase of the day. This should rule out the variability of diurnal fluctuations of melatonin secretion. The average min before surgery (time of first sampling) has been reported in Table 1.
Table 1: were data distributed normally (as mean and sd are provided), otherwise median and IQR or range are likely more accurate ?
Re: The data was analyzed with the assistance of a biostatistician. Control group has been changed in ‘untreated’ group. Figure 2 has been changed to better present the data.
It would be useful to have some more information on the oxygenation/saturation targets used during the surgery. Pre)-oxygenation has been applied, but how has oxygen been handled throughout surgery/anesthesia. Please check the fentanyl dose, should likely be microgr ?
Re: Details about the oxygen target/saturation used during surgery have been added to the method section. Oxygen was titrated to achieve a peripheral oxygen saturation of at least 92% during surgery (range 86-95%)
Please be consistent on the names used, either T1 and T2, or T0 and T1 (text versus figures) a please provide more details to the timing of sample collection related to surgery, and to dosing? There is neither information yet on the melatonin measurements (sampling, how it has been measured)
Re: Detalied informations on melatonin measurements have been added in method section. Sampling was carried out for each newborn at least one hour after administration of melatonin (T0) and upon return to intensive care after surgery (T1).
Editing issue on how the references are mentioned in the text ?
Re: The editing issue has been solved
I therefore respectfully suggest 'major = moderate' revision
Reviewer 2 Report
This is a report of a single center substudy of a multicenter trial. Infants undergoing surgery were randomized to receive melatonin or placebo preoperatively. Blood was sampled, and melatonin levels as well as markers of oxidative stress were measured pre-and post operatively. these markers were non protein bound iron, F2-isoprostanes and advanced oxidation protein products.
The administration of melatonin resulted in considerably increased melatonin levels pre- and postoperatively.
Non protein bound iron was significantly lower postoperatively compared to preoperatively in both study groups. F2 isoprostanes and advanced oxidation protein products also decreased from preoperative to postoperative in both study groups, however, the decreases were significant only in the melatonin group. This is interpreted by the authors as evidence for an anti-oxidative effect by melatonin. However, the decrease regarding F2 ioprostanes appears to be significant only because the preoperative levels where higher in the melatonin group, thus resulting in a larger decrease while postoperative levels between either group are similar. In a head-to-head comparison between control and melatonin groups, postoperative levels would probably not be different. The authors do not comment, how the higher preoperative levels of F2 isoprostane levels in the melatonin group might be interpreted.
Regarding advanced oxidation protein products, the decrease from preoperative to postoperative does not look at all that different between control and melatonin groups. Indeed, the difference in the melatonin group is a bit larger than in the control group, rendering the former significant but in a direct comparison of advanced oxidation protein product levels between both groups, no significant differences would probably be found.
Therefore, this reviewer is less enthusiastic about any positive antioxidative effects of melatonin being proven here, and would recommend a more careful interpretation of the obtained data.
Minor points: in such a report of scientific findings, sentences should usually be written in past tense. This was not always the case in this manuscript. Only the last sentence of the abstract and the last sentence of the conclusions could appropriately use the present tense.
Author Response
Reviewer 2
This is a report of a single center substudy of a multicenter trial. Infants undergoing surgery were randomized to receive melatonin or placebo preoperatively. Blood was sampled, and melatonin levels as well as markers of oxidative stress were measured pre-and post operatively. these markers were non protein bound iron, F2-isoprostanes and advanced oxidation protein products.
The administration of melatonin resulted in considerably increased melatonin levels pre- and postoperatively.
Non protein bound iron was significantly lower postoperatively compared to preoperatively in both study groups. F2 isoprostanes and advanced oxidation protein products also decreased from preoperative to postoperative in both study groups, however, the decreases were significant only in the melatonin group. This is interpreted by the authors as evidence for an anti-oxidative effect by melatonin. However, the decrease regarding F2 isoprostanes appears to be significant only because the preoperative levels where higher in the melatonin group, thus resulting in a larger decrease while postoperative levels between either group are similar. In a head-to-head comparison between control and melatonin groups, postoperative levels would probably not be different. The authors do not comment, how the higher preoperative levels of F2 isoprostane levels in the melatonin group might be interpreted.
Re: We thank the reviewer for the observation. We do not have a clear explanation for why the F2 isoprotane were higher in the pre-operative period in the melatonin group. We did randomize patients to the untreated and melatonin group and hence the surgical conditions and acuity were similar between the two groups. We therefore, can only present the results.
It might be considered that all the patients enrolled in the study, who undergo surgery, belong to a very sensitive patient population in which the "stress condition" is very marked. In fact, the levels of oxidative stress biomarkers were found to be significantly higher in both the Untreated and Mel-treated groups than those measured in healthy infants we analyzed in a previous study (see Longini et al. Mediators Inflamm 2017, 1758432). Melatonin, on the other hand, at physiological plasma concentration, tried to reduce oxidative stress during the postoperative period, which could be recognized as a physiological attempt to respond to oxidative stress, but it reached the significant effect only after exogenous administration which led to more than 100 fold-increase of melatonin plasma concentration. We are hopeful that these results highlight the positive antioxidant effects of melatonin in patients undergoing neonatal surgery.
Finally, it is also necessary to consider that this is a pilot study including a small sample in which randomization could play an important role, and that we recognized as a limitation of this study.
Regarding advanced oxidation protein products, the decrease from preoperative to postoperative does not look at all that different between control and melatonin groups. Indeed, the difference in the melatonin group is a bit larger than in the control group, rendering the former significant but in a direct comparison of advanced oxidation protein product levels between both groups, no significant differences would probably be found.
Therefore, this reviewer is less enthusiastic about any positive antioxidative effects of melatonin being proven here, and would recommend a more careful interpretation of the obtained data.
Re: Each group served as their own control with a baseline and then a change from baseline after surgery. Our data demonstrates a significant change in the melatonin group from baseline. To our group, this was the best way to analyze the data. The above comment may be true, but it does not take into account the change from baseline.
The AOPP levels were significantly reduced only after exogenous administration that resulted in very high melatonin serum concentration compared to the physiological levels. The small sample size of this study is likely to have limited the levels of significance below what might have been established in a larger sample size, that is, the study is under- powered for efficacy Information about limitations of the data and analysis, given the small sample size and pilot nature of this study, including recommendations for further larger-scale research has been outlined in the discussion section. We have not made any statement indicating that the results of this study are reliable on their own.
Minor points: in such a report of scientific findings, sentences should usually be written in past tense. This was not always the case in this manuscript. Only the last sentence of the abstract and the last sentence of the conclusions could appropriately use the present tense.
Re: Sentences have now rewritten in past tense
Round 2
Reviewer 1 Report
I suggest to accept, but would suggest the authors to check again the fentanyl dose currently suggested 10-50, as this could perhaps also be 1-5 microgr/kg ? (10-50 reads very high) at proofprint
Author Response
We thank again the reviewer for his/her time and effort. The mistake has been corrected
Reviewer 2 Report
The report has been inmproved and the conclusion is now adequate. Some points, however, remain:
Significance tests should be done between the untreated and treated groups rather than before versus after treatment.
Lines 253 -257 are redundant since the data is shown in Fig. 3. According to the results of the new testing mentioned above, the text needs to be adjusted.
Line 350 "tended", not "tried"
Author Response
Reviewer 2
The report has been improved and the conclusion is now adequate. Some points, however, remain:
Significance tests should be done between the untreated and treated groups rather than before versus after treatment.
Re: Following the reviewer suggestion, a new table has been added in the manuscript including all the data about the oxidative stress biomarkers concentrations (Table 2). The mean and trends of all measurements between Untreated and Mel groups and between the pre and post-operative period have been provided.
Lines 253 -257 are redundant since the data is shown in Fig. 3. According to the results of the new testing mentioned above, the text needs to be adjusted.
Re: Lines 253-257 have been deleted, as all results of biomarkers concentrations have been reported in Table 2
Line 350 "tended", not "tried"
Re: The word “tried” has been replaced with “tended”